# Combined Effects of Straw Return with Nitrogen Fertilizer on Leaf Ion Balance, Photosynthetic Capacity, and Rice Yield in Saline-Sodic Paddy Fields

Kun Dang [1,†], Cheng Ran [1,†], Hao Tian [1], Dapeng Gao [1], Jinmeng Mu [1], Zhenyu Zhang [1], Yanqiu Geng [1], Qiang Zhang [1,2], Xiwen Shao [1,*] and Liying Guo [1,2]

1   Agronomy College, Jinlin Agricultural University, Changchun 130118, China;
    dangkun920713@163.com (K.D.); m15754365764@163.com (C.R.); tianhao163mail@163.com (H.T.);
    gaodapeng5163@163.com (D.G.); stevenmjm@163.com (J.M.); 19398167548@163.com (Z.Z.);
    ccgyq@163.com (Y.G.); qiangz@jlau.edu.cn (Q.Z.); guoliying0621@163.com (L.G.)
2   Key Laboratory of Germplasm Innovation and Physiological Ecology of Coldland Grain Crops,
    Ministry of Education, Harbin 150000, China
*   Correspondence: shaoxiwen@126.com
†   These authors contributed equally to this work.

**Abstract:** Soil salinization is a prevalent global environmental issue that significantly hampers crop growth and yield. However, there has been limited research on the impact of nitrogen fertilization and various management practices in alleviating saline-sodic stress in crops. In order to examine the impact of combined straw and nitrogen fertilizer application on the physiological and photosynthetic characteristics of rice in saline-sodic paddy fields, a three-year field experiment was conducted in Jilin Province, China. The experiment was conducted as a split-zone trial, where the main zone consisted of straw (S) and the secondary zone consisted of nitrogen fertilizer (N). Two levels of straw were 0 t ha$^{-1}$ (B) and 7 t ha$^{-1}$ (T). Four nitrogen treatments were applied: 0, 150, 250, and 350 kg ha$^{-1}$, denoted as N0, N1, N2, and N3, respectively. The results show that the combination of straw and nitrogen fertilizer has been found to effectively reduce the Na$^+$/K$^+$ value, malondialdehyde content, and the relative electric leakage of rice leaves in saline-sodic soil. Furthermore, it increases leaf water potential, relative water content, and chlorophyll content, thereby promoting rice photosynthesis and improving rice yield. The rice yield exhibited the greatest positive effect when straw and nitrogen fertilizer were combined at a rate of 250 kg ha$^{-1}$. The effectiveness of this combination improves over time. However, it is important to avoid excessive application of nitrogen fertilizer when using straw returning. This approach not only ensures stable rice yield in saline-sodic fields, but also has positive effects on the economic impact of fertilizer application and soil environment preservation.

**Keywords:** rice; saline-sodic stress; straw; nitrogen fertilization; yield

## 1. Introduction

Soil salinization is a widespread environmental issue globally. According to Mostofa et al. (2015), approximately 831 million hectares of land are affected by saline-sodic worldwide [1]. Among the three major saline-sodic concentration areas in the world, the western Songnen Plain of China stands out, with over $3.0 \times 10^6$ ha of saline-sodic land [2]. The main soil salts in this region are NaHCO$_3$ and Na$_2$CO$_3$, and the soil pH ranges from 8.5 to 10.5 [3]. However, despite being one of the major grain-producing areas in China, the Songnen Plain faces significant limitations in crop yields, due to the presence of saline-sodic soil [4]. Saline-sodic soil has a detrimental impact on plant growth and development due to osmotic imbalance, ion toxicity, and high pH stress [5]. On the one hand, the excessive presence of salt ions in saline-sodic soil leads to a decrease in soil osmotic potential, creating osmotic stress on the plant membrane system. This stress restricts water absorption by



plant cells, thereby affecting plant growth [6]. On the other hand, saline-sodic soil with a high concentration of $Na^+$ ions can disrupt the balance of ion flux in plant tissue cells [7,8]. This disruption leads to the accumulation of MDA, which in turn causes peroxidation of the plasma membrane and the deterioration of membrane structure and function [9]. At the same time, the high pH stress of saline-sodic soil hinders the absorption of essential ions such as $Ca^{2+}$, $Mg^{2+}$, and $NO_3^-$ in crop roots. This stress also affects the synthesis of chlorophyll and inhibits photosynthesis, thereby impacting crop yield formation [10]. As soil salinization increases, it poses a serious threat to crop production. Therefore, finding ways to improve saline-sodic soil and minimize the detrimental effects of saline-sodic stress on crops is crucial for the rational development and utilization of such soil.

Nitrogen fertilizer is globally recognized as the most widely produced and utilized fertilizer, playing a pivotal role in enhancing the nutritional and reproductive growth of crops, thereby increasing crop yield [11]. Furthermore, extensive research has demonstrated that nitrogen accumulation serves as a crucial factor in enhancing crop stress resistance. (i) The efficient utilization of nitrogen can enhance the antioxidant metabolism of plants by elevating the activities of vital antioxidant enzymes and the levels of non-enzyme components [12]. (ii) The application of the appropriate amount of nitrogen can increase the content of soluble sugars and proline in leaves, resulting in an increase in osmotic pressure in the crop. This, in turn, enables the crop to mitigate the inhibitory effect of saline-sodic stress on root water uptake through osmoregulation [13]. (iii) Nitrogen also plays a role in influencing the activity of a series of enzymes involved in photosynthetic reactions and contributes to the synthesis of chlorophyll. As a result, it promotes photosynthesis in crops grown in saline-sodic land and improves crop yield [14]. Previous studies have demonstrated the positive effects of nitrogen application on crops such as maize [15], rice [16], and wheat grown in saline-sodic land [17]. Nitrogen application has been found to influence crop nitrogen uptake and metabolism [18], leading to improved saline-sodic tolerance in crops and reduced growth inhibition in saline-sodic land [19]. However, implementing proper nitrogen application in crops is challenging due to issues related to the amount and method of application [20]. The study demonstrated that insufficient nitrogen fertilizer results in a decrease in rice yield, preventing rice varieties from achieving their maximum yield potential. Additionally, it leads to excessive consumption of soil nitrogen and a decline in soil fertility [21]. Conversely, the excessive application of nitrogen fertilizer alone not only increases the number of ineffective tillers in rice and reduces the tiller tassel rate but also disrupts the hormone balance in plants, inhibiting their growth and adversely affecting rice yield formation [22]. Long-term excessive application of nitrogen fertilizer can lead to environmental pollution and ecological damage, as well as an increase in salinization and soil compaction in saline-sodic fields [23]. Therefore, finding a practical approach to nitrogen application is a pressing issue that needs to be addressed in saline-sodic land.

Straw return has gained importance as a green straw treatment method in China [24]. This method not only reduces soil bulk density but also improves soil aeration and soil aggregate quantity [22]. Additionally, straw return increases the quantity and activity of soil microbes by effectively regulating surface soil temperature, thereby enhancing biodiversity [23]. Moreover, straw has a significant positive impact on saline-sodic land. It improves the stability of soil structure, inhibits the accumulation of salt on the soil surface, and reduces the salt content in the rhizosphere soil [25]. Previous studies have demonstrated that the incorporation of straw into saline-sodic soil enhances soil porosity and aeration, thereby facilitating the absorption of nutrients and water by crops [26,27]. Zhao (2016) further revealed that placing straw layers at depths of 20 or 30 cm effectively disrupts soil capillaries, reduces the accumulation of $Na^+$ in the surface soil, and optimizes the crop growing environment [28]. Furthermore, the abundance of nutrients such as carbon, nitrogen, phosphorus, and potassium in straw contributes to the improvement of soil fertility and nutrient availability in saline-sodic soil, ultimately promoting crop growth upon its return to the field [29,30]. However, other studies have shown that straw return

has an adverse effect on crop nitrogen uptake and yield formation [31]. These studies indicate that the high carbon content in straw can significantly stimulate the growth of soil microorganisms. This stimulation leads to intense competition for nitrogen between crops and soil microorganisms. As a result, the nitrogen is immobilized through soil microbial adsorption, resulting in an insufficient supply of nitrogen fertilizer for crops and ultimately affecting crop yield formation [32]. The gradual decomposition of straw in the early stage has a negative impact on the soil structure, crop seed germination, and seedling root growth, ultimately leading to a decrease in crop yield [33]. Therefore, it is imperative to reassess the methods of straw application in saline-sodic land in order to minimize the adverse effects of straw return and ensure the sustainability of this practice.

Previous studies have shown that the combination of straw and nitrogen fertilizer is the most effective approach for maintaining soil nitrogen accumulation and balancing crop yield [34]. The simultaneous addition of straw and nitrogen fertilizer can enhance the water status and chlorophyll content of the crop, ultimately influencing its yield [35]. These findings indicate that returning straw along with appropriate nitrogen fertilizer is beneficial for crop growth and yield formation. However, the impact of straw combined with nitrogen fertilizer on the photosynthetic and physiological characteristics of rice in saline-sodic paddy fields remains uncertain.

The purpose of this 3-year study was to investigate the effects of straw combined with nitrogen fertilizer on the enhancement of photosynthetic characteristics, ion balance, leaf water status, and yield of rice in saline-sodic paddy fields. Additionally, the study aimed to explore the underlying mechanism of action. We made the following assumptions: (i) The combination of straw and nitrogen fertilizer maintained the balance of $Na^+$ and $K^+$ in rice leaves in saline-sodic paddy fields, thereby ensuring the integrity of membrane structure and function. (ii) The application of straw and nitrogen together could improve the water condition and enhance the photosynthetic performance of rice leaves in saline-sodic paddy fields. (iii) The combination of straw and nitrogen fertilizer resulted in an increase in rice yield and a reduction in the nitrogen application rate on saline-sodic paddy fields.

## 2. Materials and Methods

### 2.1. Experimental Site

Field trials were conducted in Da'an City, Jilin Province, China (N 45°35′58″–N 45°36′28″, E 123°50′27″–123°51′31″), from April 2017 to October 2019. The region experiences a typical mid-temperate monsoon climate with a transitional zone from a semi-humid to semi-arid climate. The spring season is characterized by dry and windy conditions, while the summer season is hot and rainy. Autumn is cool, and the short winter is dry and cold, with minimal snowfall. The average annual rainfall in the area is 399.9 mm, with the majority of rainfall occurring from June to August. The annual evaporation rate is 1840 mm, which is 4.5 times higher than the annual rainfall. The area experiences an average of 2915 h of sunshine per year, with an average temperature of 5.2 °C. The accumulated temperature over the year is 2996.2 °C, and the frost-free period lasts for 144 days. Figure 1 illustrates the average precipitation and temperature during the test year. The table below (Table 1) provides the basic physical and chemical properties of the soil (0–20 cm) before the test. According to the World Soil Resources Reference Basis [36], the soil type is classified as Solonetz.

**Table 1.** Basic physical and chemical properties of the tested soil.

| Parameter | Unit | Mean | Parameter | Unit | Mean |
|-----------|------|------|-----------|------|------|
| Bulk density | $g\ cm^{-3}$ | 1.61 | Total N | $g\ kg^{-1}$ | 0.27 |
| pH | - | 10.10 | Total K | $g\ kg^{-1}$ | 22.36 |
| Salinity | $g\ kg^{-1}$ | 4.78 | Organic matter | $g\ kg^{-1}$ | 7.31 |
| $ENa^+$ | $cmol_c\ kg^{-1}$ | 3.21 | Alkalihydrolysis N | $mg\ kg^{-1}$ | 16.3 |
| CEC | $cmol_c\ kg^{-1}$ | 12.97 | Available P | $mg\ kg^{-1}$ | 16.90 |
| ESP | % | 24.74 | Available K | $mg\ kg^{-1}$ | 107.25 |

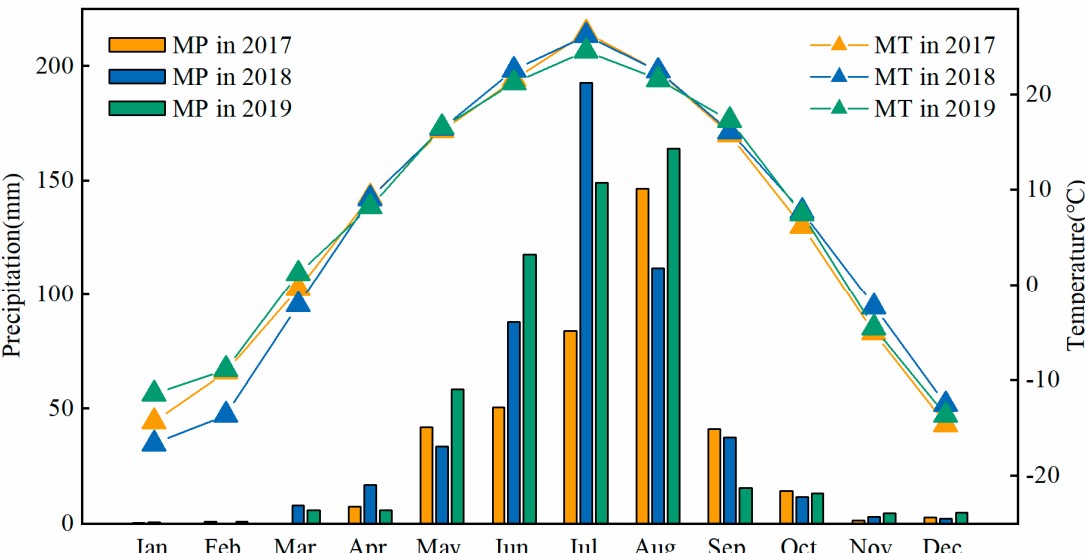

**Figure 1.** Variations in monthly precipitation and temperature in 2017, 2018 and 2019 years within the study area. MP: Monthly precipitation; MT: Monthly temperature.

*2.2. Experimental Design*

The experiment followed a split-plot trial design, with straw (S) as the main zone and nitrogen fertilizer (N) as the secondary zone. Two levels of straw were considered: 0 t ha$^{-1}$ (B) and 7 t ha$^{-1}$ (T), where the total incorporation was determined based on the local rice grain-straw ratio of 7 t ha$^{-1}$. Nitrogen treatments of 0, 150, 250, and 350 kg ha$^{-1}$ were denoted as N0, N1, N2, and N3, respectively, and each treatment was replicated three times. A total of 24 plots, each covering an area of 36 m$^2$ (8 m × 4.5 m), were established for this experiment. The plots were separated by ridges measuring 50 cm in width and 30 cm in height. To minimize the impact of fertilizer between treatments, each plot had an independent single row and a single irrigation system. The nitrogen fertilizer used was urea, with a nitrogen content of 46%. Nitrogen fertilizer was applied at a ratio of 6:3:1 for base fertilizer, mid-tillering fertilizer, and panicle fertilizer, respectively. The superphosphate fertilizer used had an application amount of 50 kg ha$^{-1}$, which was applied as a one-time base fertilizer. The fertilizer has a P$_2$O$_5$ content of 16%. The potassium fertilizer was applied at a rate of 75 kg ha$^{-1}$, following a ratio of 6:4 for base fertilizer and panicle fertilizer, respectively, with a K$_2$O content of 60%.

In this study, a straw returning test was conducted in the same plot for three consecutive years. The straw used in the previous season was manually collected from a nearby farmland. After air-drying the straw under natural conditions, it was cut into 5–7 cm pieces using a straw chopper. In spring, the field water layer was controlled to 1–2 cm through beating, and the straw was evenly spread on the field surface. A small beating machine was employed to mix the straw with the 0–20 cm soil uniformly, ensuring a consistent and even distribution of the straw's impact when returned to the field. Additionally, base fertilizer was applied simultaneously with the straw return, mid-tillering fertilizer was applied in the middle and late June (around 20 June), and panicle fertilizer was applied in mid-July (around 15 July).

The rice varieties were selected from the control variety, Changbai 9, which is known for its tolerance to saline-sodic land. Transplanting was conducted around 20 May, with a planting density of 20 hills and a spacing of 16.5 × 30.0 cm, with five seedlings per hill. The harvest took place around 1 October for three consecutive years. After transplanting, the paddy fields were flooded, and a floodwater depth of 3–5 cm was maintained until two weeks prior to harvest to prevent any loss of biomass and yield. Strict control measures were implemented to manage insects, diseases, and weeds throughout the entire growth period of the rice.

*2.3. Sampling Methods and Measurements*

2.3.1. Determination of $Na^+$ and $K^+$ Content in Rice Leaves

During the heading stage of rice, three rice plants exhibiting similar growth patterns were chosen from each plot. The selected rice leaves were dried at 105 °C for 0.5 h and then at 80 °C for 48 h. After being ground into a fine powder and screened, a 0.5 g accurately weighed sample was heated and digested with $H_2SO_4$-$H_2O_2$. The content of $Na^+$ and $K^+$ in the sample was determined using a flame photometer (M410, Sherwood Scientific Ltd., Cambridge, UK). The power supply and gasoline combustion pump were turned on, and the intake speed was adjusted to create a jagged flame after ignition. The boiled sample was then tested and the reading recorded. The concentration of $Na^+$ and $K^+$ in the sample was calculated based on the drawn standard curve [37].

2.3.2. Determination of Malondialdehyde (MDA) Content and Relative Electrolyte Leakage (REL), Superoxide (SOD) and Peroxidase (POD) Activities in Rice Leaves

During the heading stage of rice, three rice plants exhibiting similar growth patterns were chosen from each plot. The contents of MDA, REL, SOD, and POD were determined using the topmost whole explants leaves. The content of MDA was determined via thiobarbituric acid staining [38]. Rice leaves (1.000 g) were weighed and ground before being mixed with 10% trichloroacetic acid (TCA, 10 mL) to create a homogenate. The homogenate was then centrifuged at $6000\times g$ for 20 min. Next, 1 mL of the supernatant was added to 2 mL of the reaction solution and boiled for 15 min. The reaction liquid consisted of 0.6% ($v/v$) thiobarbituric acid (TBA) and 10% ($w/v$) trichloroacetic acid. The mixture was cooling centrifuged again at $4000\times g$ for 15 min. The supernatant was collected and the absorbance values at 450, 532, and 600 nm were measured. The level of lipid peroxidation was quantified in nanomoles per gram of fresh weight, using an extinction coefficient of $155 \text{ mM}^{-1}\text{cm}^{-1}$.

The REL of rice leaves was determined according to Dionisio-Sese and Tobita [39]. Fresh leaves weighing 1.000 g were washed with deionized water and then transferred to a test tube containing 15 mL of deionized water. The leaves were incubated at room temperature (25 °C) for 2 h, and the conductivity (E1) was measured using a conductivity meter (DS-307, Shanghai Reitz, Shanghai, China). Subsequently, the test tube was placed in a 100 °C environment for 30 min and then cooled back to room temperature (25 °C) to measure the electrical conductivity (E2). The relative electrolyte leakage rate (REL) was calculated using the following formula:

$$REL(\%) = \frac{E1}{E2} \times 100 \qquad (1)$$

The determination of superoxide dismutase (SOD) was carried out using the method described by Jini and Joseph [9]. Fresh leaves weighing 0.500 g were homogenized in 5 mL of potassium phosphate buffer (100 mM, pH 7.8). The buffer consisted of ethylenediamine tetraacetic acid (EDTA, 0.1 mM), Triton X-100 (2.4 mM), and polyethylpyrrolidone (0.2 μm). The supernatant was obtained through filtration and centrifugation ($15,000\times g$, 15 min) at 4 °C. Subsequently, the reaction solution was prepared by combining sodium bicarbonate-sodium buffer (50 mM, pH 9.8), EDTA (0.1 mM), epinephrine (0.6 mM, added last), and enzymes (3.0 mL). Absorbance values were measured at 475 nm over a period of 4 min.

Peroxidase (POD) activity was determined following the method described by Assaha et al. Fresh leaves (0.100 g) were homogenized in 3 mL of phosphate buffer (0.1 M, pH 7.0). The resulting homogenate was then centrifuged at 4 °C ($18,000\times g$) for 15 min. The substrate used was O-diphenylamine (1 mg mL$^{-1}$ in methanol). The oxidation of o-diphenylamine was measured at 430 nm. The reaction mixture consisted of phosphoric acid buffer (0.1 M, pH 6.5), freshly prepared o-diphenylamine solution (0.1 mL), and enzyme extract. To initiate the reaction, 0.2 mL of $H_2O_2$ (0.2 M) was added and mixed. The change in absorbance per minute at 420 nm was recorded.

### 2.3.3. Determination of Leaf Relative Water Content (RWC) and Water Potential ($\Psi_w$)

During the heading stage of rice, three rice plants exhibiting similar growth patterns were chosen from each plot. The RWC and $\Psi_w$ of leaves were then determined. The $\Psi_w$ of the first unfurling leaf of rice (1 leaf per plant, 3 leaves per treatment) was measured between 4 h and 6 h using an HR-33T dew point microvoltmeter (Wescor Inc., Logan, UT, USA) [16]. The relative water content was determined using the methods described by Machado and Paulsen [40]. The initial step involved determining the fresh weight (FW) of rice leaves. Subsequently, the leaves were immersed in clean water within a sealed plastic bag for a duration of 24 h. Afterward, the surface water was removed by blotting with absorber paper, and the weight of the leaves in their saturated state (SW) was measured. The dry weight (DW) was then determined by drying the leaves in an oven at a temperature of 80°C for a period of 48 h. The formula used for calculating the leaf relative water content is as follows:

$$RWC = (FW - DW)/(SW - DW) \times 100\% \tag{2}$$

### 2.3.4. Determination of Photosynthetic Parameters and SPAD Value of Rice Leaves

At the heading stage of rice, three rice plants exhibiting similar growth trends were chosen for each treatment in order to measure the SPAD value and photosynthetic parameters of rice flag leaves. The net photosynthetic rate (A), stomatal conductance (gs), and transpiration rate (E) of flag leaves were measured via Li-6400 (Li-Cor Inc., Lincoln, NE, USA) photosynthetic apparatus from 9:00 am to 11:00 am on a clear day without wind. Each treatment was repeated five times to ensure accuracy. The temperature of the leaf chamber was maintained at approximately 26 °C, while the light intensity was set at 800 $\mu mol \cdot m^{-2} \cdot s^{-1}$. Furthermore, the $CO_2$ concentration was kept at 400 $\mu mol \cdot mol^{-1}$, and the relative humidity ranged between 60% to 70% during the measurements. The chlorophyll content of flag leaves (leaves used to determine photosynthetic parameters) was measured with a SPAD-502 portable chlorophyll meter (Minolta, Osaka, Japan).

### 2.3.5. Determination of Rice Yield

During the rice maturity stage, we randomly selected three sites from each treated plot. At each site, we harvested a 3 m$^2$ area to determine the yield per unit area (t ha$^{-1}$). The rice yield at harvest time was adjusted to 14% grain water content.

### 2.3.6. Statistical Analysis

All data were collected and analyzed using Microsoft Excel 2019 software. The data were then analyzed using the SPSS statistical package version 22 (IBMCorp., Armonk, NY, USA). Descriptive statistics were used to test the mean and standard error of measurement parameters. The mutual influence of straw-returning treatment and nitrogen application level on the measured parameters was determined through a two-factor analysis of variance. Simulated linear and quadratic regression analyses were conducted to examine the relationships between MDA and Na$^+$/K$^+$, REL and MDA, yield and SPAD, yield and RWC, and yield and nitrogen. The Duncan multiple comparison method was utilized to assess the statistical significance of the difference ($p < 0.05$). The results are presented as standard error (SE). Graphs were created using Origin 2021 software.

## 3. Results

### 3.1. Na$^+$ and K$^+$ Contents and the Na$^+$/K$^+$ Ratio of Rice Leaves

The results of this study demonstrated that straw returning combined with nitrogen fertilizer had a significant positive impact on the Na$^+$ content, K$^+$ concentration, and Na$^+$/K$^+$ ratio of rice in saline-sodic land. Figure 2 illustrates that, under the same nitrogen fertilizer level, the straw-returning treatment led to a decrease in Na$^+$ concentration and Na$^+$/K$^+$ ratio, while increasing the K$^+$ concentration. Across all nitrogen levels, there was a decrease of 20.5%, 20.1%, and 32.7% in Na$^+$ concentration, and an increase of 35.1%, 31.5%, and 46.2% in K$^+$ concentration, respectively, when compared to the absence of straw

returning from 2017 to 2019. Furthermore, the $Na^+/K^+$ ratio decreased by 40.1%, 39.1%, and 53.5%, respectively.

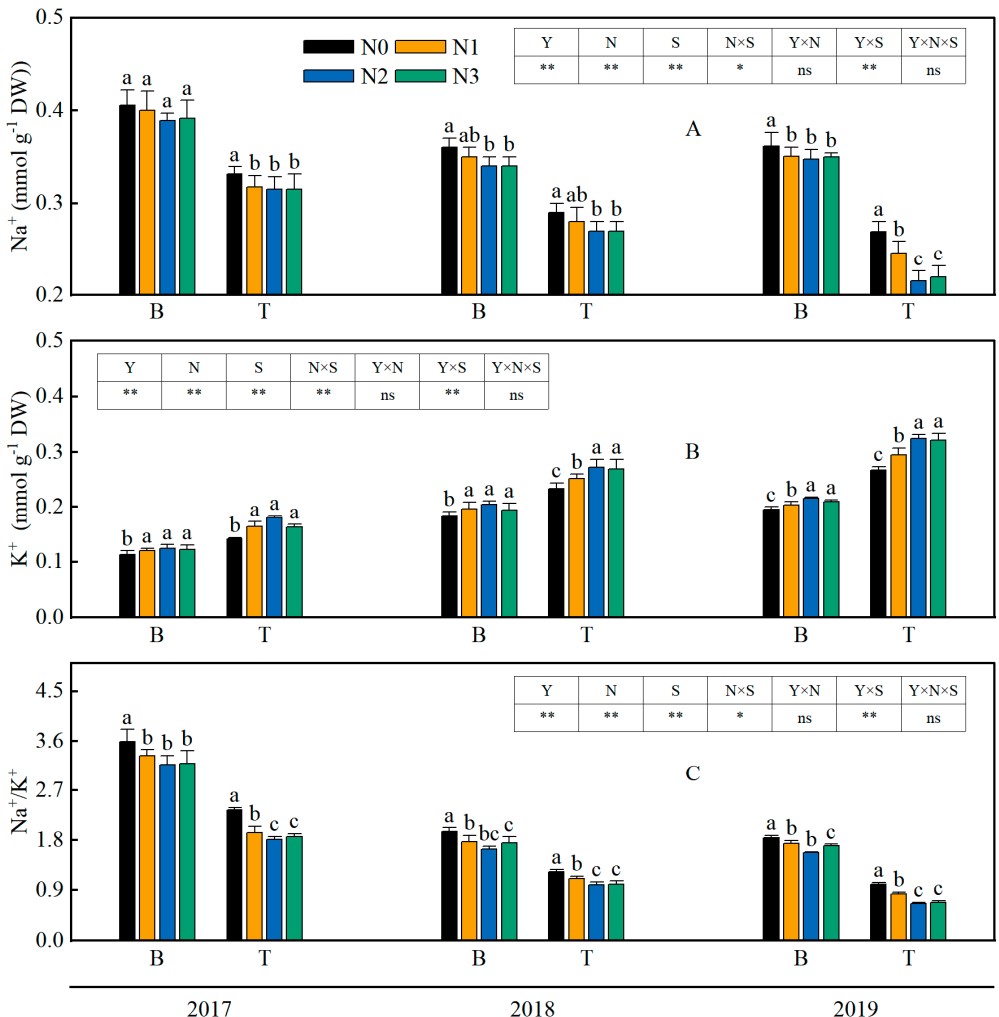

**Figure 2.** The effects of straw nitrogen fertilizer on the ratio of $Na^+$ (**A**), $K^+$ (**B**), and $Na^+/K^+$ (**C**) in rice leaves in saline-sodic fields from 2017 to 2019. The mean values of three repetitions ± SE ($n$ = 3) were used, and different letters were used to indicate statistical significance at the $p < 0.05$ level. ns: non-significant, *: significant at $p < 0.05$, **: significant at $p < 0.01$. B: no straw returning to field; T: 7 t ha$^{-1}$ of straw returning to field. N0, N1, N2 and N3 were 0, 150, 250 and 350 kg ha$^{-1}$ of nitrogen fertilizer, respectively. Y: year, N: nitrogen fertilizer, S: straw.

Under the condition of returning straw to the field, the concentration of $Na^+$ and the $Na^+/K^+$ ratio initially decreased and then increased with the increase in the nitrogen application rate. On the other hand, the concentration of $K^+$ initially increased and then decreased with the increase in the nitrogen application rate. In 2019, under the N2 treatment, the $Na^+$ concentration and $Na^+/K^+$ ratio reached their lowest levels, while the $K^+$ concentration reached its maximum. Specifically, the $Na^+$ concentration in the N2 treatment was significantly lower (20.0% and 12.4%) compared to the N0 and N1 treatments, respectively. Similarly, the $Na^+/K^+$ ratio was significantly lower (34.1% and 20.3%) in the N2 treatment compared to the N0 and N1 treatments. On the other hand, the $K^+$ concentration significantly increased (17.7% and 9.0%) in the N0 and N1 treatments. There were no significant differences in $Na^+$, $K^+$, and $Na^+/K^+$ ratios between the N2 and N3 treatments. Under the same level of nitrogen fertilizer, the soil $Na^+$ concentration and $Na^+/K^+$ ratio decreased, while the $K^+$ concentration increased with the increase in straw-returning years.

### 3.2. Malondialdehyde (MDA) and Relative Electricity Leakage (REL) of Rice Leaves

As depicted in Figure 3, it was observed that under identical nitrogen fertilizer levels, the act of straw returning led to a decrease in the MDA content and REL of rice. When straw was present, the MDA content of rice in saline-sodic land followed the order N0 > N1 > N3 > N2, while the REL of rice in saline-sodic land followed the order N0 > N1 > N2 > N3. Notably, the MDA content of the N2 treatment in 2019 was significantly lower by 8.6% and 2.3% compared to the N0 and N1 treatments, respectively. Similarly, the REL of the N2 treatment was also significantly lower (reduced by 5.9% compared to the N0 treatments). There was no significant difference in the MDA content and REL of rice between N2 treatment and N3 treatment in saline-sodic land. When considering the same nitrogen fertilizer level, it was observed that the MDA content and REL of rice leaves decreased with the increase in the number of years of straw returning. In comparison to the N2 treatment in 2017, the MDA content and REL of the N2 treatment in 2019 decreased by 14.1% and 8.6%, respectively. It is worth noting that the MDA content and REL also exhibited significant responses to the interaction between Y × S.

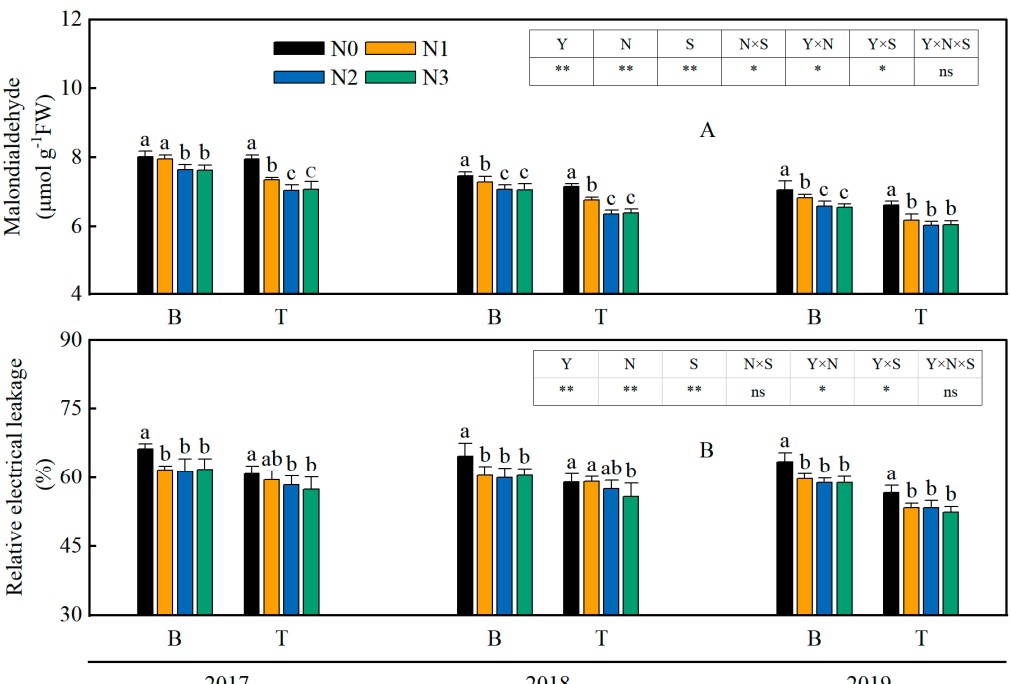

**Figure 3.** The effects of straw nitrogen fertilizer on the ratio of malondialdehyde (MDA) (**A**) and relative electricity leakage (REL) (**B**) in rice leaves in saline-sodic fields from 2017 to 2019. The mean values of three repetitions ± SE (*n* = 3) were used, and different letters were used to indicate statistical significance at the *p* < 0.05 level. ns: non-significant, *: significant at *p* < 0.05, **: significant at *p* < 0.01. B: no straw returning to field; T: 7 t ha$^{-1}$ of straw returning to field. N0, N1, N2, and N3 were 0, 150, 250, and 350 kg ha$^{-1}$ of nitrogen fertilizer, respectively. Y: year, N: nitrogen fertilizer, S: straw.

The study (Figure 4) investigated the correlation between MDA and Na$^+$/K$^+$ as well as REL and MDA in rice leaves in saline-sodic land from 2017 to 2019. The results revealed a significant positive correlation (*p* < 0.01) between MDA and Na$^+$/K$^+$ as well as REL and MDA in rice leaves in saline-sodic soil. The correlation coefficients were found to be 0.74 and 0.79, respectively, under the no-straw-returning treatment, and 0.90 and 0.64 under the straw-returning treatment. These findings suggest that the MDA content in rice leaves is closely associated with the Na$^+$/K$^+$ content and that the MDA content in rice leaves in saline-sodic soil has a significant impact on REL.

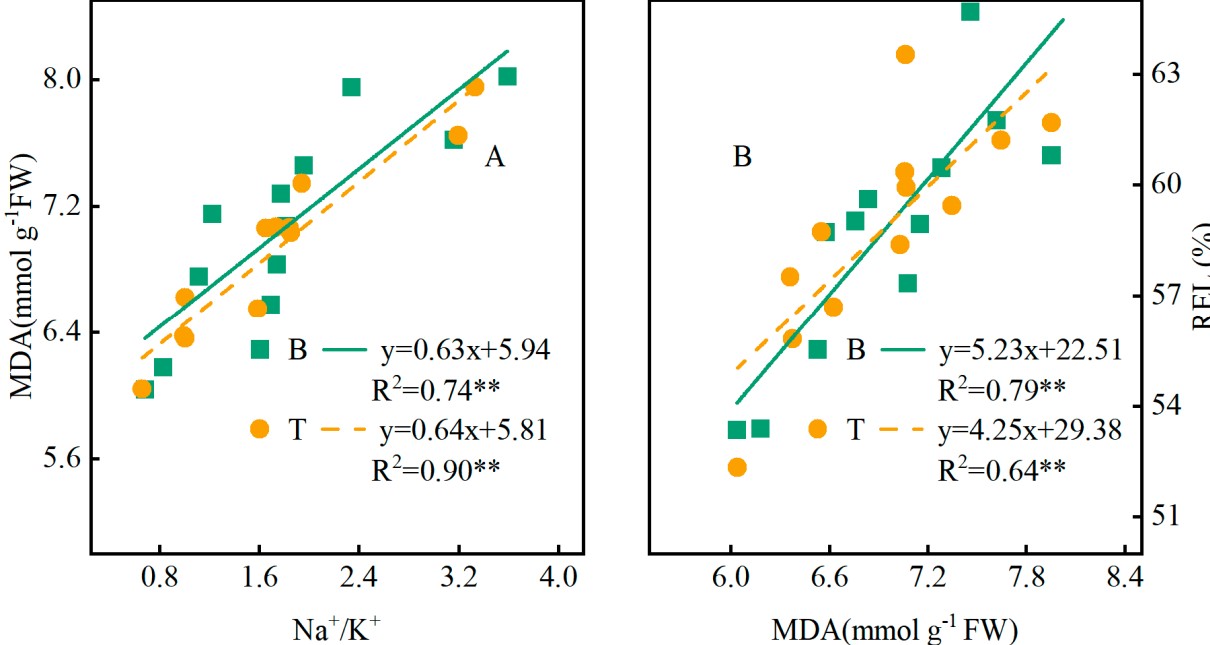

**Figure 4.** Relationship between rice MDA and $Na^+/K^+$ (**A**), REL and MDA (**B**) in 2017–2019. B: no straw returning to field; T: 7 t ha$^{-1}$ of straw returning to field. ** is significantly correlated with $p < 0.01$. MDA: malondialdehyde; REL: relative electricity leakage.

### 3.3. Superoxide Dismutase (SOD) and Peroxidase (POD) Activities of Rice Leaves

The activities of SOD and POD in rice were significantly reduced when straw was returned under the same nitrogen fertilizer level in saline-sodic land (Figure 5). With straw reflux treatment, the activities of SOD and POD decreased as the nitrogen application rate increased. When straw was returned, the N2 treatment showed a significant decrease in SOD activity of rice of 17.7% and 12.2%, and a significant decrease in POD activity of rice of 15.2% and 9.3% compared to the N0 and N1 treatments. There was no significant difference in the activities of SOD and POD between the N2 and N3 treatments. The activities of SOD and POD decreased as the number of years of straw return increased under the same nitrogen fertilizer level.

### 3.4. Relative Water Content (RWC) and Leaf Water Potential ($\Psi_w$) of Rice Leaves

Both nitrogen fertilizer and straw returning had a significant impact on rice RWC and $\Psi_w$ ($p < 0.01$). The results showed that under the straw-returning treatment in 2019, the RWC and $\Psi_w$ of N2 treatment were the highest, which were 60.83% and 1.21 MPa, respectively (Figure 6). Under the same nitrogen fertilizer level, the RWC and $\Psi_w$ of the straw-returning treatment were higher compared to the no-straw-returning treatment. Both the RWC and $\Psi_w$ of the straw-returning treatment and the no-straw-returning treatment showed a trend of initially increasing and then decreasing with the increase in nitrogen application rate. Among them, the RWC of the N2 treatment in 2019 was significantly higher than that of the N0 and N1 treatments by 11.8% and 4.4%, respectively, and the $\Psi_w$ of the N2 treatment was also significantly higher than that of the N0 and N1 treatments by 7.1% and 4.5%, respectively. There was no significant difference in the MDA content and REL between the N2 and N3 treatments. Under the same nitrogen fertilizer level, both the RWC and $\Psi_w$ increased with the increase in straw-returning years. The results are presented in Figure 6, and it can be observed that the interaction between S × Y has a noteworthy impact on the RWC and $\Psi_w$ of rice grown in saline-sodic soil.

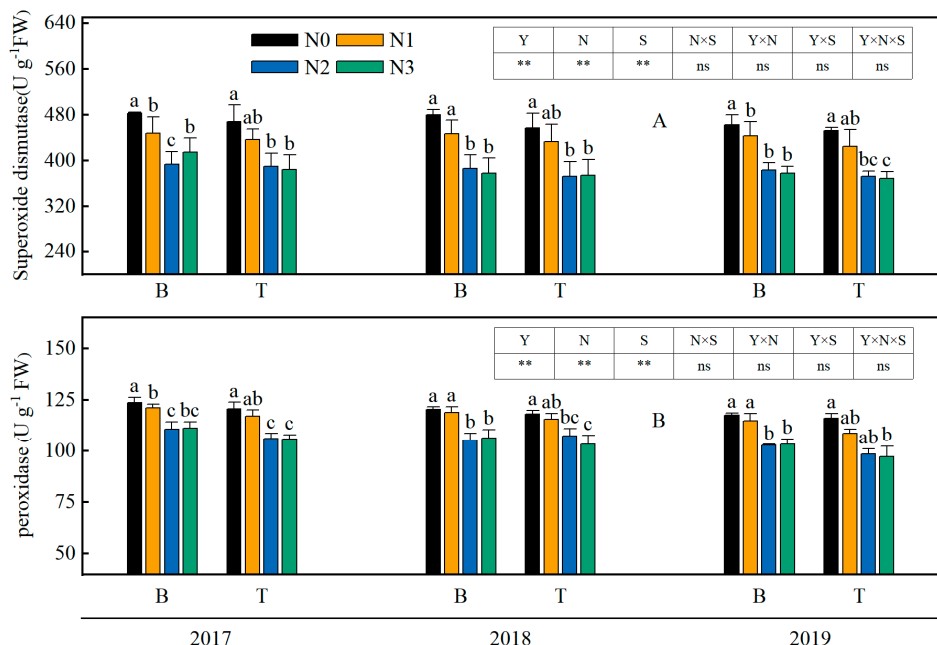

**Figure 5.** The effects of straw nitrogen fertilizer on the ratio of superoxide dismutase (SOD) (**A**) and peroxidase (POD) (**B**) in rice leaves in saline-sodic fields from 2017 to 2019. The mean values of three repetitions $\pm$ SE (*n* = 3) were used, and different letters were used to indicate statistical significance at the *p* < 0.05 level. ns: non-significant, **: significant at *p* < 0.01. B: no straw returning to field; T: 7 t ha$^{-1}$ of straw returning to field. N0, N1, N2, and N3 were 0, 150, 250, and 350 kg ha$^{-1}$ of nitrogen fertilizer, respectively. Y: year, N: nitrogen fertilizer, S: straw.

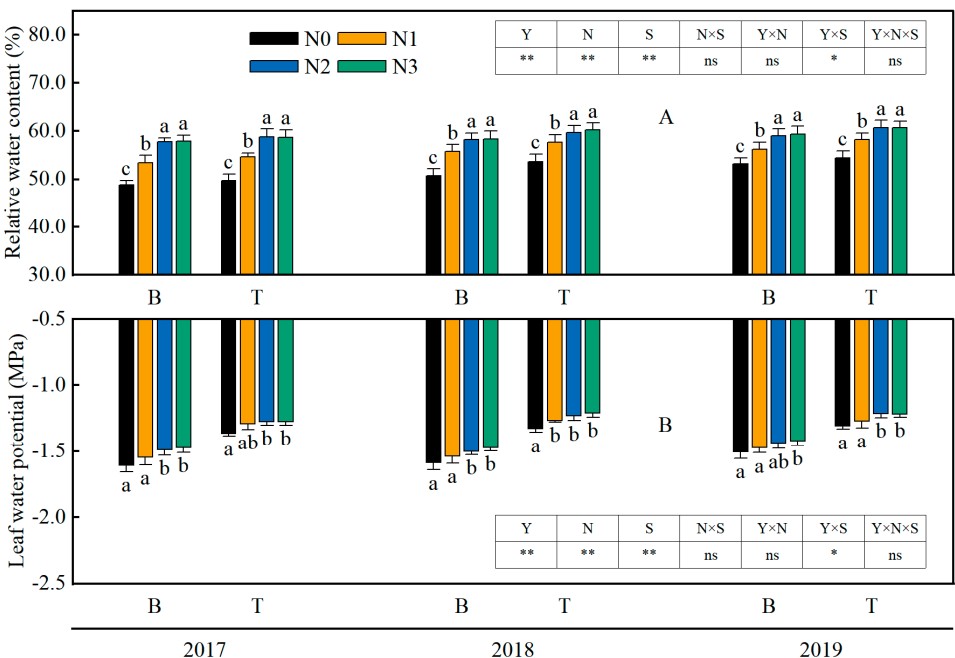

**Figure 6.** The effects of straw nitrogen fertilizer on relative water content (RWC) (**A**) and leaf water potential ($\Psi_w$) (**B**) in rice leaves in saline-sodic fields from 2017 to 2019. The mean values of three repetitions $\pm$ SE (*n* = 3) were used, and different letters were used to indicate statistical significance at the *p* < 0.05 level. ns: non-significant, *: significant at *p* < 0.05, **: significant at *p* < 0.01. B: no straw returning to field; T: 7 t ha$^{-1}$ of straw returning to field. N0, N1, N2, and N3 were 0, 150, 250, and 350 kg ha$^{-1}$ of nitrogen fertilizer, respectively. Y: year, N: nitrogen fertilizer, S: straw.

### 3.5. SPAD Value of Rice Leaves

From 2017 to 2019, the SPAD values of rice leaves in saline-sodic soil exhibited significant responses to nitrogen and straw fertilization. As shown in Figure 7A, under the same nitrogen fertilizer level, the SPAD value of straw-returning treatment was 6.0–28.9% higher than that of no-straw-returning treatment. SPAD values increased with increasing N application for both straw return and no-straw-returning treatments. In the straw returning condition, the SPAD value of the N2 treatment in 2019 was significantly higher than that of the N0 and N1 treatments by 34.4% and 9.7%, respectively. There was no significant difference between the N2 and N3 treatments. The SPAD value increased with the increase in the year of straw return under the same nitrogen fertilizer level.

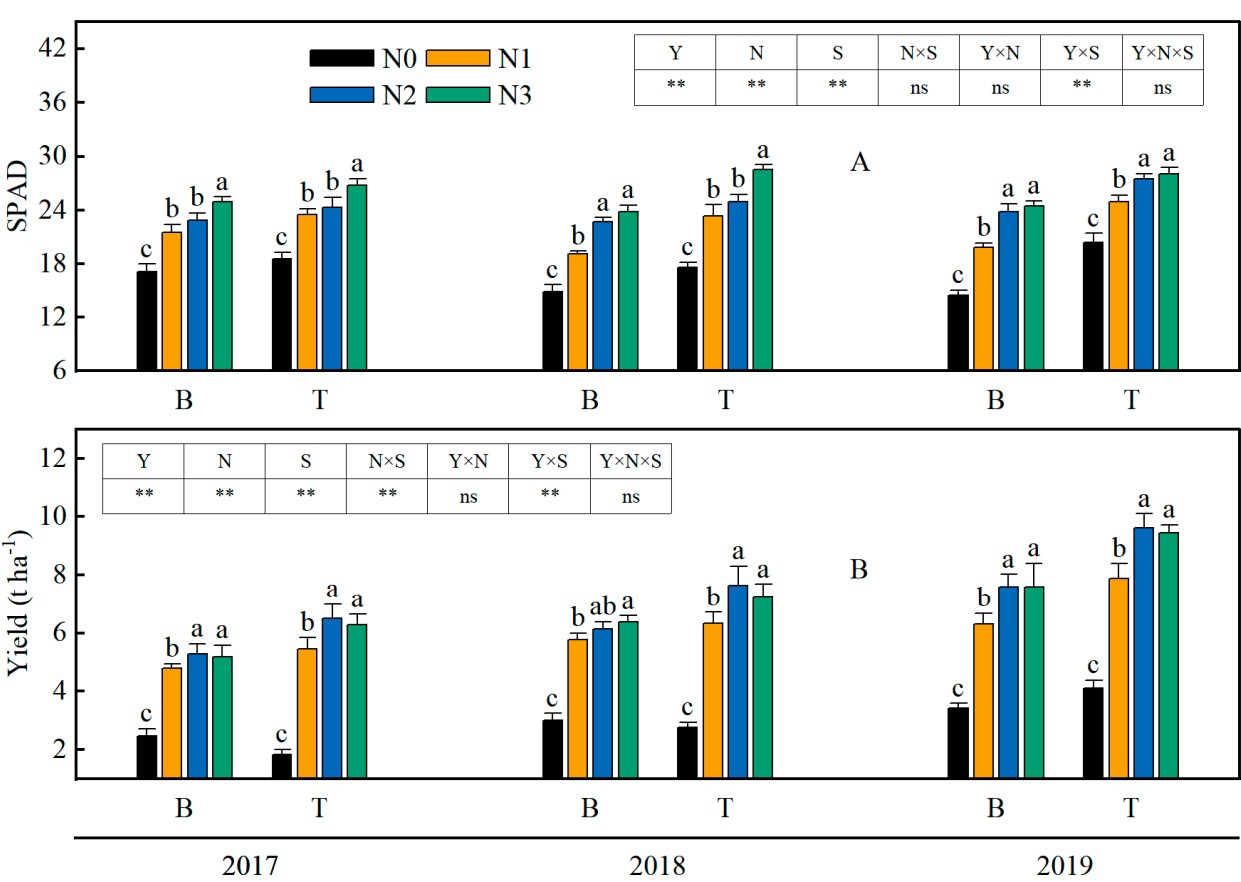

**Figure 7.** The effects of straw nitrogen fertilizer on SPAD in rice leaves (**A**) and yield (**B**) in saline-sodic fields from 2017 to 2019. The mean values of three repetitions ± SE (*n* = 3) were used, and different letters were used to indicate statistical significance at the *p* < 0.05 level. ns: non-significant, **: significant at *p* < 0.01. B: no straw returning to field; T: 7 t ha$^{-1}$ of straw returning to field. N0, N1, N2, and N3 were 0, 150, 250, and 350 kg ha$^{-1}$ of nitrogen fertilizer, respectively. Y: year, N: nitrogen fertilizer, S: straw.

### 3.6. Net Photosynthetic Rate (A), Stomatal Conductance (gs), and Transpiration Rate (E) of Rice

From 2017 to 2019, the rice A, gs, and E were significantly influenced by nitrogen fertilizer (N), straw returning (S), and their interaction (N × S) under saline-sodic stress (Figure 8). It was observed that, when compared to the absence of straw-returning treatment, the rice leaves exhibited higher A, gs, and E levels under the straw-returning treatment at the same nitrogen fertilizer level (Figure 8). The levels of A, gs, and E in rice leaves showed an initial increase and then a decrease with the increase in nitrogen application, both with and without straw returning. In 2019, under the condition of straw returning to the field, the A, gs, and E values of the N2 treatment were significantly higher than those of the N0 and N1 treatments. Specifically, A was 21.8% and 15.3% higher, gs was

13.2% and 6.0% higher, and E was 12.9% and 11.0% higher, respectively. There was no significant difference in SPAD values between the N2 and N3 treatments. Under the same nitrogen fertilizer level, the values of A, gs, and E in rice leaves increased as the years of straw returning increased. This indicates that the interaction between straw and returning year (Y × S) significantly influenced A, gs, and E, suggesting that straw returning to the field had a cumulative effect.

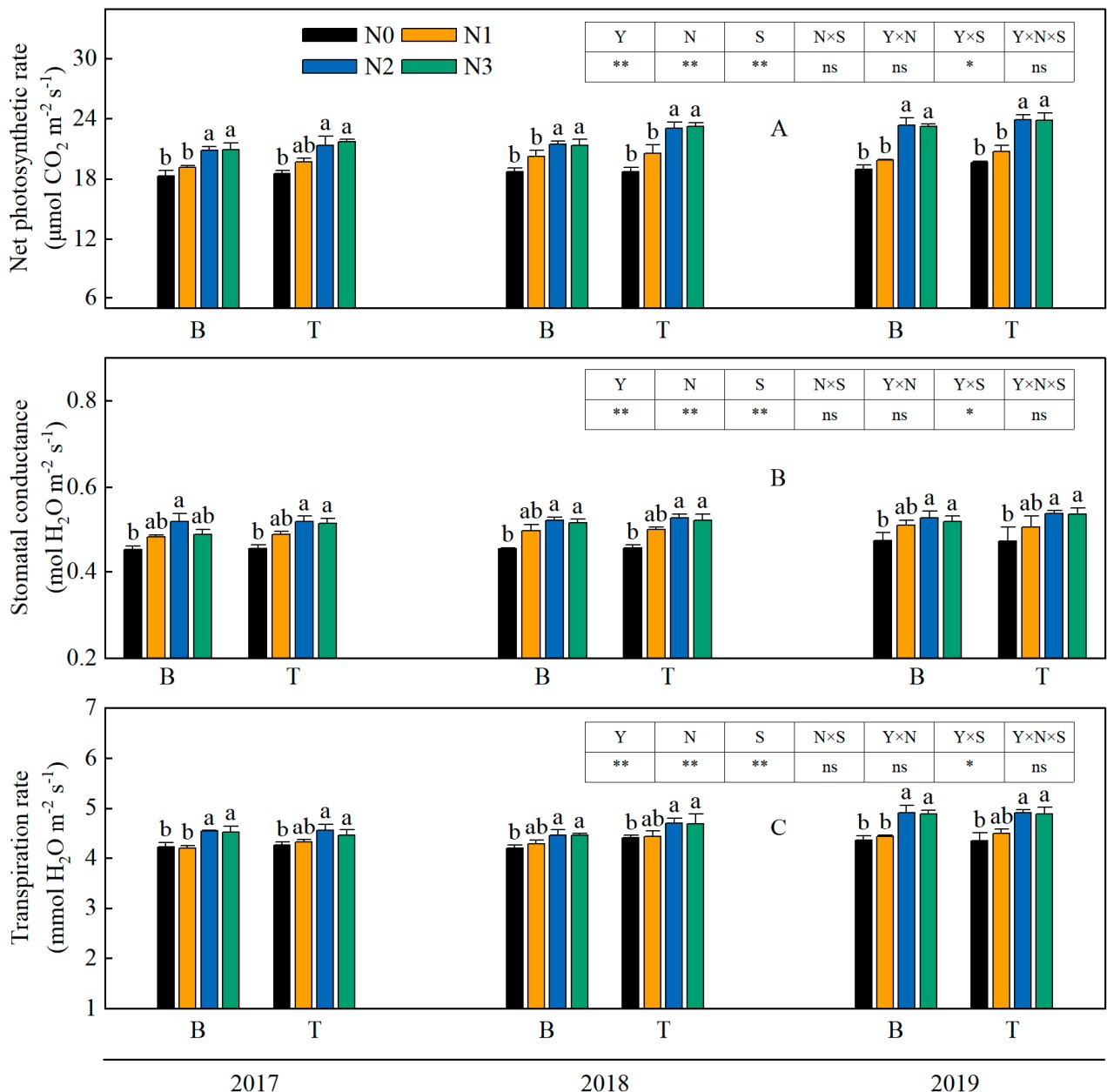

**Figure 8.** The effects of straw nitrogen fertilizer on net photosynthetic rate (**A**), stomatal conductance (gs) (**B**) and transpiration rate (E) (**C**) in rice leaves in saline-sodic fields from 2017 to 2019. The mean values of three repetitions ± SE (*n* = 3) were used, and different letters were used to indicate statistical significance at the *p* < 0.05 level. ns: non-significant, *: significant at *p* < 0.05, **: significant at *p* < 0.01. B: no straw returning to field; T: 7 t ha$^{-1}$ of straw returning to field. N0, N1, N2, and N3 were 0, 150, 250, and 350 kg ha$^{-1}$ of nitrogen fertilizer, respectively. Y: year, N: nitrogen fertilizer, S: straw.

### 3.7. Grain Yield of Rice

From 2017 to 2019, the application of nitrogen fertilizer and straw, as well as their interaction (N × S), were found to significantly affect rice yield in saline-sodic soil ($p < 0.01$). As depicted in Figure 7B, the highest rice yield in 2019 was observed in the N2 treatment, reaching 9.64 t ha$^{-1}$ (Figure 7B). Under the same nitrogen fertilizer level, the rice yield in the straw-returning treatment was 9.8–27% higher than that in the no-straw-returning treatment, except for the N0 treatment. In the case of N0 treatment, the yield of the straw-returning treatment was 26.0% and 8.6% lower than that of the no-straw-returning treatment in 2017 and 2018, respectively. According to studies conducted in 2019, it was observed that rice yields experienced an increase of 20.4%, 25.3%, 27.0%, and 24.3% with the implementation of the straw-returning treatment. With the increase in nitrogen application rate, rice yield initially increased and then decreased in both the straw-returning and no-straw-returning treatments. The rice yield of the N2 treatment was significantly higher than that of the N0 and N1 treatments, but there was no significant difference between the N3 and N2 treatments. Under the same nitrogen fertilizer level, the rice yield in saline-sodic soil increased as the years of straw returning increased. The results in Figure 7B demonstrate that the interaction between straw-returning years (Y) and nitrogen fertilizer level (S) has a significant influence on rice yield in saline-sodic soil. This suggests that straw returning to the field has a cumulative effect.

In addition, this study (Figure 9) investigated the relationship between yield and SPAD, as well as yield and RWC, in rice leaves from saline-sodic soil during the period of 2017–2019. The findings revealed a significant positive correlation between leaf water content and leaf SPAD on rice yield in saline-sodic soil ($p < 0.01$). The correlation coefficients for the non-straw-returning treatment were 0.68 and 0.86, while those for the straw-returning treatment were 0.78 and 0.86, respectively. These results suggest a close association between the water content and SPAD value of rice leaves with rice yield.

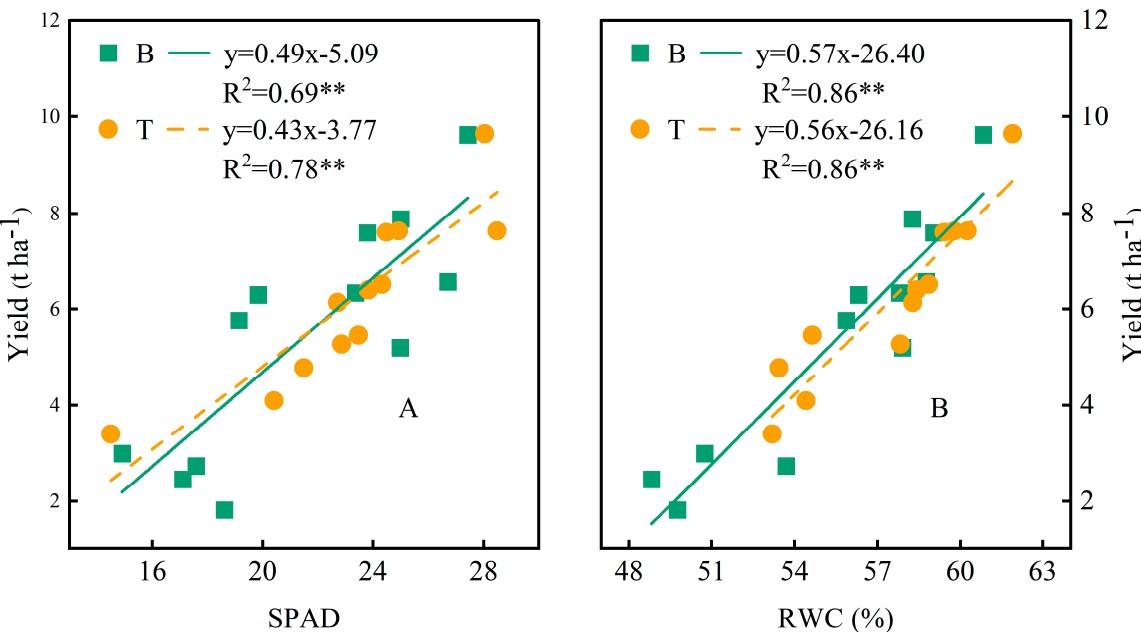

**Figure 9.** Relationship between rice yield and SPAD (**A**), yield and RWC (**B**) in 2017–2019. B: no straw returning to field; T: 7 t ha$^{-1}$ of straw returning to field. ** is significantly correlated with $p < 0.01$. Yield: rice yield; RWC: Relative water content.

The relationship between rice yield and N fertilizer application before and after straw addition was explored by correlation analysis (Figure 10). A highly significant quadratic correlation ($p < 0.01$) was found between rice yield and nitrogen application from 2017–2019. The correlation coefficients under no straw return conditions were 0.99, 0.99, and 0.99,

respectively. The correlation coefficients were 0.99, 0.97, and 0.99 under the straw-returned condition, respectively. Nitrogen reduction refers to the difference between the amount of nitrogen fertilizer used when straw is added to achieve the maximum rice yield without straw and the amount of nitrogen fertilizer used when the maximum rice yield without straw is added. According to the regression analysis, the reduction in N fertilizer use from 2017 to 2019 was 49.3%, 50.4%, and 60.8%, respectively. The regression analysis also showed that under straw-returning conditions, the maximum theoretical yields (peak of the curve) from 2017 to 2019 were 6.53, 7.54, and 9.56 t ha$^{-1}$, respectively, corresponding to 283.73, 284.69, and 307.45 t ha$^{-1}$ of nitrogen fertilizer application, respectively. This indicates that the optimum N fertilizer application rate varied with increasing years of straw return.

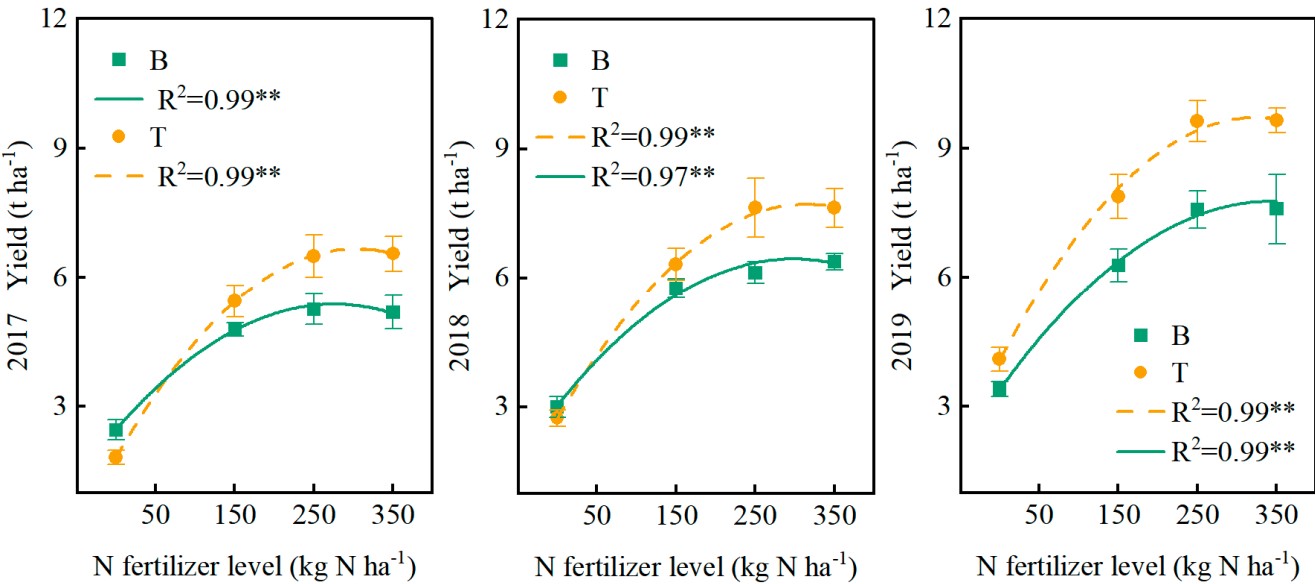

**Figure 10.** Reduction of nitrogen fertilizer application in saline-sodic paddy under straw returning to field from 2017 to 2019. B: no straw returning to field; T: 7 t ha$^{-1}$ of straw returning to field. ** is significant correlation at $p < 0.01$.

## 4. Discussion

### 4.1. The Application of Both Straw and Nitrogen Fertilizer Effectively Maintained the Ion Balance of Rice Tissue in Saline-Sodic Land

Saline-sodic conditions have a significant impact on plant growth and development, primarily due to osmotic imbalance, ion toxicity, and high pH stress [5]. In such conditions, the soil's high concentration of Na$^+$ results in excessive absorption of Na$^+$ by plants, inhibiting the uptake of essential nutrient ions like K$^+$ and NO$_3^-$. This interference negatively affects the growth, development, and even survival of plants. Additionally, the high Na$^+$ concentration can displace Ca$^{2+}$ in the cell membrane system, causing damage to the membrane structure and compromising its integrity and function. Consequently, intracellular K$^+$ and other nutrient ions as well as organic solutes are excreted, disrupting the ion balance within plant cells [41]. In this study, it was found that straw return reduced the Na$^+$ concentration of rice (Figure 2A). This reduction may be attributed to several factors: (i) The long-term application of straw increased the content of cations such as Ca$^{2+}$, Mg$^{2+}$, and NH$_4^+$ in the soil of saline-sodic land. This, in turn, improved the cation exchange capacity of rice root cell membranes and reduced the uptake of Na$^+$ by rice roots [42]. (ii) The decomposition of straw produces colloids that have an adsorbent effect on Na$^+$, thereby preventing its uptake by the rice root system [43]. (iii) Long-term straw return also resulted in changes to the structure of soil aggregates. This, in turn, reduced the diffusion of Na$^+$ from deep soil to the cultivated layer, optimized the rhizosphere soil of rice, and ultimately decreased the concentration of Na$^+$ in rice tissues [44]. In addition, the combined application of straw and nitrogen fertilizer has been found to be more effective

when the straw is returned to the field. This is because nitrogen can enhance the activity of soil microorganisms and promote the decomposition of straw by these microorganisms. As a result, the soil's carbon cycle is improved, cation exchange capacity and soil structure are enhanced, and the uptake of $Na^+$ by rice roots in the soil is reduced [25]. Additionally, nitrogen has been found to increase the expression of transporter proteins, which helps prevent the uptake of $Na^+$ by the root system of rice [12]. The entry of $Na^+$ into plant root cells primarily occurs through two pathways: non-selective cation channels and high-affinity $K^+$ channels [45]. Under saline-sodic stress, the similar hydration radius of $Na^+$ and $K^+$ leads to a competition between a large amount of $Na^+$ and $K^+$ channels, resulting in the inhibition of $K^+$ absorption [41,46]. However, the combination of straw return and nitrogen fertilizer application limits the absorption of $Na^+$ and reduces its concentration in rice tissues (Figure 2A), while increasing the absorption of $K^+$ (Figure 2B). Furthermore, potassium is an essential plant nutrient, and an increased concentration of $K^+$ in the soil is considered a key mechanism to counteract saline-sodic stress and promote crop growth [38]. The decomposition of straw increased the content of $K^+$ in the soil and promoted the uptake of $K^+$ by rice roots, which is crucial for plant saline-sodic resistance [47]. Therefore, the combination of straw and nitrogen fertilizer reduced the $Na^+/K^+$ ratio in rice tissue, resulting in an improvement in the saline-sodic resistance of rice (Figure 2C).

### 4.2. The Combined Application of Straw and Nitrogen Fertilizer Enhanced the Oxidative Damage and Water Status of Rice Leaves in Saline-Sodic Land

The content of malondialdehyde (MDA) is an indicator of the extent of damage to the plant cell membrane [48]. When plants are exposed to saline-sodic stress, the excessive accumulation of $Na^+$ in their tissues disrupts the integrity and function of the membrane structure. This disruption causes an imbalance in reactive oxygen species (ROS) and leads to an increase in the MDA content, which is a byproduct of membrane lipid peroxidation [49]. In this study, the combined application of straw and nitrogen fertilizer had a positive impact on reducing the MDA content of rice leaves in saline-sodic paddy fields (Figure 3A). This effect could be attributed to the reduction in $Na^+$ concentration in rice leaves, as demonstrated in Figure 4A, which showed a significant positive correlation with MDA content.

The relative electricity leakage (REL) of tissue is an important indicator for assessing cell membrane integrity. A higher REL in plant tissue indicates greater damage to the cell membrane. The study showed that the higher the MDA content of plant tissues, the greater the REL of the tissues [50], which was consistent with the results of this study. As shown in Figure 4B, the MDA content of rice was significantly positively correlated with its REL. The combined application of straw and nitrogen fertilizer enhanced the saline-sodic resistance of rice, inhibited MDA production, and reduced REL, thereby ensuring cell membrane integrity (Figure 3B).

Under saline-sodic stress, reactive oxygen species (ROS) also serves as a stress signal, triggering the activation of antioxidant enzymes like superoxide dismutase (SOD) and peroxidase (POD) to maintain the equilibrium of ROS in stressed cells [51]. Previous studies have highlighted the significance of SOD and POD in the antioxidant enzyme system, with SOD being primarily responsible for eliminating $O_2^-$ followed by POD [9,52]. In our study, we observed an increase in the activities of SOD and POD enzymes under saline-sodic stress (Figure 5). However, when straw and nitrogen fertilizer were applied together, the resistance of rice to saline-sodic stress improved, leading to a balanced ROS level and subsequently reducing the activities of SOD and POD enzymes.

Under saline-sodic stress, the accumulation of harmful ions such as $Na^+$ in plant leaves significantly increases the ion content in plant cells and the permeability of the cell membrane. This makes it difficult for cells to absorb water, resulting in a decrease in leaf water potential ($\Psi_w$) [53]. In this study, the combined application of straw and nitrogen fertilizer reduced the concentration of $Na^+$ and the content of MDA in the rice tissues in a saline-sodic rice area. This application also maintained cell integrity and permeability,

improved water absorption capacity, and increased $\Psi_w$ and RWC of the leaves (Figure 6). Additionally, the application of nitrogen fertilizer may enhance the accumulation of amino acids in plant tissues. Amino acids serve as osmotic protectants, offsetting the increase in osmotic potential of saline-sodic soil and increasing the RWC and $\Psi_w$ of leaves [54].

*4.3. The Application of Straw and Nitrogen Fertilizer in Combination Resulted in an Increase in the SPAD Value and Photosynthetic Capacity of Rice Leaves in Saline-Sodic Land*

Under saline-sodic stress, the excessive accumulation of $Na^+$ in plant leaves disrupts the structure and function of membranes, resulting in an imbalance of reactive oxygen species (ROS) and an increase in membrane lipid peroxidation products. This damage extends to the chloroplast membrane, ultimately leading to chlorophyll degradation [55,56]. The SPAD value serves as an indicator of chlorophyll content [57]. In this study, the combined application of straw and nitrogen fertilizer significantly enhanced the SPAD values of rice in saline-sodic paddy fields (Figure 7A). This positive effect can be attributed to the combined application of straw and nitrogen fertilizer, which effectively reduces the $Na^+$ content in plant leaves, maintains membrane integrity, prevents chlorophyll degradation, and creates a favorable environment for chlorophyll synthesis, ensuring smooth progress in the synthesis process. Furthermore, the decrease in chlorophyll content under saline-sodic stress may also be attributed to reduced nitrogen uptake [58]. However, the combined application of straw and nitrogen fertilizer promotes rice root vigor and increases nitrogen content in the leaves, consequently boosting leaf chlorophyll content [59,60].

Photosynthesis serves as the foundation for plant growth and development. It enables plants to convert light energy into stable chemical energy and transform inorganic matter from the environment into organic matter. This process provides essential material and energy for plant physiological metabolism, as well as growth and development [61]. Saline-sodic stress has been found to impede the synthesis of photosynthetic pigments, resulting in a reduction of photosynthetic characteristics in leaves [10]. Additionally, saline-sodic stress can hinder the absorption of nitrogen by plants, which is closely associated with the synthesis of photosynthetic enzymes. Consequently, it can inhibit the production of photosynthase and impact its activity [62,63]. This study demonstrates that the combined application of straw and nitrogen in saline-sodic paddy fields leads to an increase in the net photosynthetic rate, stomatal conductance, and transpiration rate of rice (Figure 8). Under stress, plants must maintain a high concentration of $K^+$ in epidermal cells and guard cells to regulate stomatal opening and closing. The combined use of straw and nitrogen fertilizer has been shown to increase the $K^+$ concentration in rice tissue and alleviate stomatal closure caused by saline-sodic stress [64]. This promotes the improvement of net photosynthetic rate and transpiration rate in plants. It is also possible that the combined application of straw and nitrogen fertilizer reduces the content of $Na^+$ in plant tissues and helps maintain membrane integrity (Figure 2A). This not only ensures smooth chlorophyll synthesis but also provides a favorable environment for efficient photosynthesis. Furthermore, nitrogen addition increases the content of photosynthetic pigments, enhancing the absorption of light energy, electron transfer, and utilization efficiency of pigments, ultimately improving the photosynthetic capacity of plant leaves [65].

*4.4. Combined Application of Straw and Nitrogen Fertilizer Improved the Rice Yield in Saline-Sodic Land*

Saline-sodic soil not only restricts photosynthesis and water use efficiency in plants, but also induces physiological drought and ion toxicity, leading to a decrease in agricultural productivity and yield [3]. In this study, the application of straw and nitrogen fertilizer in combination resulted in an increase in rice yield in saline-sodic soil. Figure 7B demonstrates a highly significant interaction between N × S and yield. The mechanism underlying the improvement of rice yield in saline-sodic paddy fields through the combined use of straw and nitrogen fertilizer is illustrated in Figure 11: (1) The combined application of straw and nitrogen fertilizer effectively reduced the $Na^+/K^+$ value (Figure 2), maintained the ion balance of rice tissues, reduced MDA content and REL (Figure 3), and ensured the integrity

of the cell membrane [66]. (2) In the present study, a significant positive correlation was found between rice yield and RWC (Figure 9B), indicating that the combination of straw and nitrogen fertilizers improved the water status of the rice, reduced osmotic stress, and increased the rice yield. (3) The combined application of straw and nitrogen fertilizer increased the SPAD value (Figure 7A), which promoted photosynthesis of rice and further improved the rice yield. The positive correlation between rice yield and SPAD value is shown in Figure 9A. Furthermore, the decay of straw releases important macro and micro nutrients, such as Ca, K, N, P, and Zn, which promote rice growth and increase rice yield [66]. This study also found that rice yield increased with the number of years of straw return (Figure 7B), indicating a cumulative effect of straw return. However, it is important to note that straw decomposition requires a nutrient supply. In the absence of N fertilizer application, straw can compete with the crop for N, resulting in an inadequate N supply to the crop [32,66]. This issue can be alleviated with the combined application of N fertilizer (Figure 7B). Furthermore, the yield of the N3 (350 kg ha$^{-1}$ N) treatment was lower than that of N2 (250 kg ha$^{-1}$ N), and the rice yield increased and then decreased with the amount of nitrogen fertilizer applied. Excessive N application can lead to greying and late maturity of rice, extensive collapse, and increased susceptibility to pests and diseases, ultimately reducing rice yield [21].

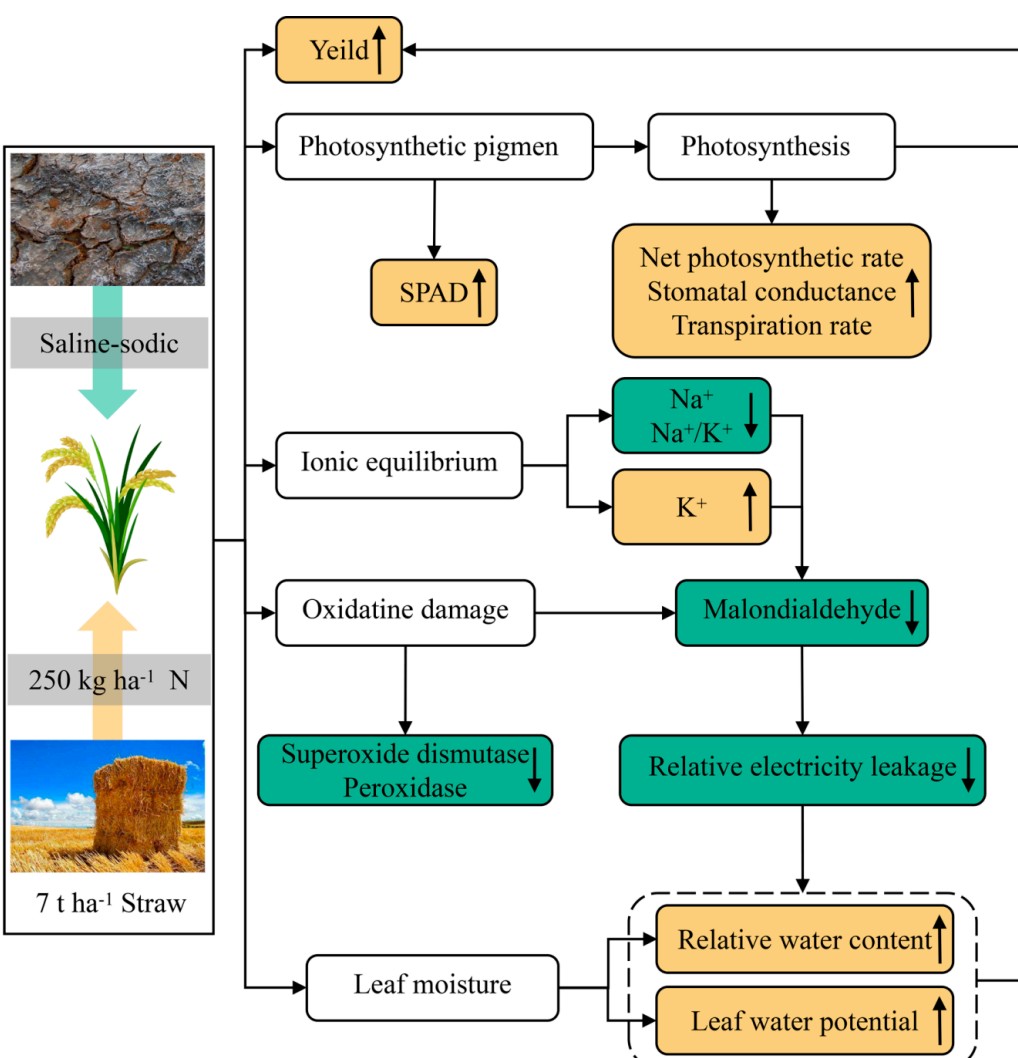

**Figure 11.** Effects of straw combined with nitrogen fertilizer on the growth and development of rice in saline-sodic paddy. The upward and downward arrows indicate the increase and decrease in each indicator.

The correlation analysis between rice yield and nitrogen application before and after straw addition was conducted (Figure 10). The results showed that straw addition could reduce the need for nitrogen application in saline-sodic paddy fields from 2017 to 2019. This reduction may be attributed to the ability of straw returning to maintain soil nitrogen storage and create a favorable soil chemical, physical, and biological environment [67,68]. Additionally, straw returning can enhance nitrogen use efficiency and decrease reliance on chemical fertilizers [68]. The optimal nitrogen fertilizer application rate in saline-sodic paddy fields under straw-returning conditions varied with the number of returning years during 2017–2019. Therefore, it is crucial to conduct a long-term study on straw and nitrogen application in saline-sodic soil to further investigate the optimal nitrogen fertilizer amount, as well as its impact on nitrogen utilization rate and the environment in saline-sodic paddy fields.

## 5. Conclusions

The results of the 3-year study demonstrated that the combined application of straw and nitrogen fertilizer effectively reduced the $Na^+/K^+$ value of rice in saline-sodic paddy fields. This application also maintained the ion balance of rice tissue, reduced the content of malondialdehyde and relative electricity leakage. As a result, it improved the water potential and relative water content of leaves, alleviating the stress damage of rice in saline-sodic paddy fields. Furthermore, the combined application of straw and nitrogen fertilizer increased the SPAD value of rice leaves, promoted the photosynthesis of rice, and ultimately led to an increase in rice yield. Therefore, The combination of straw returning and nitrogen fertilizer has been found to be beneficial for the growth and yield of rice. It has been observed that the effectiveness of this combination improves with time. In this study, it was found that the best positive effect on yield was achieved when straw was combined with 250 kg ha$^{-1}$ nitrogen fertilizer. It is important to avoid excessive application of nitrogen fertilizer when straw is returned to the field. This method not only ensures the stability of rice yield in saline fields but also has positive effects on the economic aspects of fertilization and soil environmental protection.

**Author Contributions:** K.D.: Conceptualization, Methodology, Software, Data curation, Writing—original draft preparation. C.R., D.G., H.T., J.M. and Z.Z.: Conceptualization, Methodology, Visualization, Investigation. Q.Z., Y.G. and L.G.: Reviewing and Editing, Supervision. X.S.: Reviewing and Editing. All authors have read and agreed to the published version of the manuscript.

**Funding:** This study was supported by National key research and development program (2022YFD1500501), Open Project of the Key Laboratory of Germplasm Innovation and Physiological Ecology of Coldland Grain Crops, Ministry of Education (CXSTOP202202).

**Data Availability Statement:** The data that support this study will be shared upon reasonable request to the corresponding author.

**Acknowledgments:** We thank the anonymous referees for their comments and suggestions that led to the improvement of this manuscript.

**Conflicts of Interest:** The authors declare no conflict of interest.

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
