# Peer review of "Combined Effects of Straw Return with Nitrogen Fertilizer on Leaf Ion Balance, Photosynthetic Capacity, and Rice Yield in Saline-Sodic Paddy Fields"

_agronomy, doi:10.3390/agronomy13092274_

Round 1

Reviewer 1 Report

The manuscript describes a study of the problem of straw return with nitrogen fertilizer on a number of physiological parameters of rice on the example of one region of China. The problems of saline-sodic soil are really important for agriculture not only in the region studied by the authors, but also in other regions where similar soils also exist. In my opinion, the research has not a very high level of novelty, but it can be useful in agronomy.

In general, I believe that such an article can be published. However, I see many inaccuracies in the description of methods, presentation of results, and formulation of conclusions.I have the following questions and comments:

1)      Line 30 & 33: “saline-sodi” - saline-sodic

2)      Materials and methods, Experimental site: It is very good that the authors provide geographic coordinates in the text, but I strongly recommend adding a map of the study of the region and fields. It is better to do this in the form of a map with insets. Since the study is regional, it is necessary to clearly show readers where the study took place.

3)      The soil characteristics in Table 1 are from some reference source, as I understand it. It is necessary to indicate from which source these characteristics are taken.

4)      Materials and methods, Sampling Methods and Measurements: methods are described very superficially or methods are not specified at all. Laboratory methods determination of Na+ and K+ content in rice leaves, determination of malondialdehyde (MDA) content and relative electrolyte leakage (REL), Superoxide (SOD) and peroxidase (POD) activities in rice leaves must be described in detail.

5)      Materials and methods, Statistical analysis: why did the authors use parametric statistical tests? It needs to be substantiated. Why were Pearson correlation analysis and regression analysis used simultaneously? Correlation analysis is a special case of regression analysis, why are both needed at the same time? At the end of the description of the results, the authors mention a little about the use of these analyzes, but this should be clearly stated in the M&M.

6)      Materials and methods, Statistical analysis: What kind of regression analysis did the authors use?

7)      Results: It is not clear what statistical methods the authors used to identify the effects of straw nitrogen fertilizer on a particular parameter. An analysis of variance was used to test the difference between the means of the two samples "N: nitrogen fertilizer" and "S: straw"? It's not quite an effect, it's an observed mean difference tested with ANOVA. Again, the question arises why the parametric test was used. And it is not entirely clear what the size of each sample is. In M&M, it was stated that the total number of samples was 24. It turns out that each sample included 12? This is too little.

8)      Column charts, which are predominantly used by the authors, represent mean values with an error of the mean. They do not reflect any of the effects of straw nitrogen fertilizer. Line graphs, perhaps filled with different colors, would look better.

9)      Lines 612-614: the regression analysis did not include this, and it is not entirely correct to draw such a conclusion based on a three-year follow-up. Although intuitively this seems to be true, the results obtained are not enough for this conclusion.

10)   In the discussion, the authors often mention the effect on cell membranes. But this is based on data from other studies. When this is put forward as an assumption, it is good, but as a conclusion (lines 605-606) it is not correct - the authors did not conduct such an experiment.

11)   Attention is drawn to the representation of the sources (references) used by the authors. It is worth expanding the range of sources, and include more from other regions.

Author Response

  • Line 30 & 33: “saline-sodi” - saline-sodic

Response: Thank you for your kindly suggestion. I have revised the text ('saline-sodi' → 'saline-sodic'), which is highlighted in line 29 of the text after modification.

  • Materials and methods, Experimental site: It is very good that the authors provide geographic coordinates in the text, but I strongly recommend adding a map of the study of the region and fields. It is better to do this in the form of a map with insets. Since the study is regional, it is necessary to clearly show readers where the study took place.

Response: Thank you for providing valuable comments. This study utilizes precise geographic coordinates to accurately depict the region (refer to lines 132-133). Your suggestion serves as a reminder of the significance of assessing the accuracy of the region. Consequently, detailed descriptions and modifications regarding the climatic conditions and soil characteristics of the region are also included (see lines 132-145) for further elaboration.

  • The soil characteristics in Table 1 are from some reference source, as I understand it. It is necessary to indicate from which source these characteristics are taken.

Response: Thank you for your kindly suggestion. we have completed the modification. While changing "According to the classification standards of the United States Department of Agriculture (1954), the soil type at this test site was determined to be high-pH saline-sodic soil. As a result, this test site can be considered representative of the hydrogeological and climatic conditions in the region." → “According to the World Soil Resources Reference Basis (IUSS Working Group, 2014), the soil type is classified as Solonetz.”

  • Materials and methods, Sampling Methods and Measurements: methods are described very superficially or methods are not specified at all. Laboratory methods determination of Na+and K+ content in rice leaves, determination of malondialdehyde (MDA) content and relative electrolyte leakage (REL), Superoxide (SOD) and peroxidase (POD) activities in rice leaves must be described in detail.

Response: Thanks for your useful comments. we have completed the detailed description of Na+ and K+ content in rice leaves, (MDA) content and relative electrolyte leakage (REL), Superoxide (SOD) and peroxidase (POD) activities in rice leaves determination method. The specific modifications made are as follows:

“At the heading stage of rice, three rice leaves were selected from each plot. From each plot, three rice leaves with the same growth trend were chosen. The selected rice leaves were then oven-dried at 105℃ for 0.5 h, followed by further oven-drying at 80℃ for 48 h. Afterward, the rice leaves were ground into a fine powder and screened. Next, 0.5 g of accurately weighed samples were heated and digested with H2SO4-H2O2. The Na+ and K+ contents in the leaves were determined using flame photometry. (Chen et al. 2021).” → “During the heading stage of rice, three rice plants exhibiting similar growth patterns were chosen from each plot. The selected rice leaves were dried at 105℃ for 0.5 h and then at 80℃ for 48 h. After being ground into a fine powder and screened, a 0.5 g accurately weighed sample was heated and digested with H2SO4-H2O2. The content of Na+ and K+ in the sample was determined using a flame photometer (M410, Sherwood Scientific Ltd., Cambridge, England). The power supply and gasoline combustion pump were turned on, and the intake speed was adjusted to create a jagged flame after ignition. The boiled sample was then tested and the reading recorded. The concentration of Na+ and K+ in the sample was calculated based on the drawn standard curve [36].”

“At the heading stage of rice, three spots were selected from each plot. From each spot, three rice plants with the same growth trend were chosen. The contents of MDA, REL, SOD, and POD were determined using the topmost whole explants leaves. The content of MDA was determined by thiobarbituric acid staining(Assaha et al. 2017). The REL of rice leaves was determined according to Dionisio-Sese and Tobita 1998. The determination of superoxide dismutase (SOD) and peroxidase (POD) was carried out using the method described by Jini and Joseph (2017).” → “During the heading stage of rice, three rice plants exhibiting similar growth patterns were chosen from each plot. The contents of MDA, REL, SOD, and POD were determined using the topmost whole explants leaves. The content of MDA was determined by thiobarbituric acid staining[37]. Rice leaves (1.000g) were weighed and ground before being mixed with 10% trichloroacetic acid (TCA, 10 ml) to create a homogenate. The homogenate was then centrifuged at 6000×g for 20 minutes. Next, 1 ml of the supernatant was added to 2 ml of the reaction solution and boiled for 15 minutes. The reaction liquid consisted of 0.6%(v/v) thiobarbituric acid (TBA) and 10%(w/v) trichloroacetic acid. The mixture was cooling centrifuged again at 4000×g for 15 minutes. The supernatant was collected and the absorbance values at 450, 532, and 600 nm were measured. The level of lipid peroxidation was quantified in nanomoles per gram of fresh weight, using an extinction coefficient of 155mM -1cm-1.

 The REL of rice leaves was determined according to Dionisio-Sese and Tobita[38]. Fresh leaves weighing 1.000 g were washed with deionized water and then transferred to a test tube containing 15 ml of deionized water. The leaves were incubated at room temperature (25℃) for 2 hours, and the conductivity (E1) was measured using a conductivity meter (DS-307, Shanghai Reitz, China). Subsequently, the test tube was placed in a 100 °C environment for 30 minutes and then cooled back to room temperature (25°C) to measure the electrical conductivity (E2). The relative electrolyte leakage rate (REL) was calculated using the following formula:

         REL(%)=E1/E2 * 100       (1)

 The determination of superoxide dismutase (SOD) was carried out using the method described by Jini and Joseph[9]. Fresh leaves weighing 0.500g were homogenized in 5 ml of potassium phosphate buffer (100 mM, pH 7.8). The buffer consisted of ethylenediamine tetraacetic acid (EDTA, 0.1 mM), Triton X-100 (2.4 mM), and polyethylpyrrolidone (0.2 μm). The supernatant was obtained through filtration and centrifugation (15000×g, 15 min) at 4°C. Subsequently, the reaction solution was prepared by combining sodium bicarbonate-sodium buffer (50 mM, pH 9.8), EDTA (0.1 mM), epinephrine (0.6 mM, added last), and enzymes (3.0 ml). Absorbance values were measured at 475 nm over a period of 4 minutes.

Peroxidase (POD) activity was determined following the method described by Assaha et al. Fresh leaves (0.100 g) were homogenized in 3 ml of phosphate buffer (0.1 M, pH 7.0). The resulting homogenate was then centrifuged at 4℃ (18000×g) for 15 min. The substrate used was O-diphenylamine (1 mg ml-1 in methanol). The oxidation of o-diphenylamine was measured at 430 nm. The reaction mixture consisted of phosphoric acid buffer (0.1 M, pH 6.5), freshly prepared o-diphenylamine solution (0.1 ml), and enzyme extract. To initiate the reaction, 0.2 ml of H2O2 (0.2 M) was added and mixed. The change in absorbance per minute at 420 nm was recorded.”

  • Materials and methods, Statistical analysis: why did the authors use parametric statistical tests? It needs to be substantiated. Why were Pearson correlation analysis and regression analysis used simultaneously? Correlation analysis is a special case of regression analysis, why are both needed at the same time? At the end of the description of the results, the authors mention a little about the use of these analyzes, but this should be clearly stated in the M&M.

Response: Thanks for the reviewers' warm tips. The reason for using a parametric statistical test is to assess the normality and homogeneity of variance of the data before conducting an analysis of variance. If the data satisfies the assumptions of normality and homogeneity of variance, ANOVA can be employed. In this study, the data meets the requirements of normal distribution and homogeneity of variance, allowing for the use of ANOVA.

Why were Pearson correlation analysis and regression analysis used simultaneously? The relationship between MDA and Na+/K+, REL and MDA, yield and SPAD, and yield and RWC were assessed by simulating linearity. Additionally, the relationship between yield and nitrogen was analyzed using quadratic regression.

We apologize for the confusion caused by the unclear expression in the text. In response to the feedback from reviewers and editors, we have made detailed revisions to the paper. The specific modifications are outlined below:“All data were collected and analyzed using Microsoft Excel 2019. Statistical analyses, including analysis of variance (ANOVA), Pearson correlation analysis, and regression analysis, were performed using SPSS (IBM Corporation, USA). The least significant difference (LSD) test was used for mean comparison, with a significance level of p < 0.05. The results are presented as standard error (SE). Graphs were created using Origin 2021 software.” → “All data were collected and analyzed using Microsoft Excel 2019. The data were then analyzed using the SPSS statistical package version 22 (IBM Corporation, USA). Descriptive statistics were used to test the mean and standard error of measurement parameters. The mutual influence of straw returning treatment and nitrogen application level on the measured parameters was determined through a two-factor analysis of variance. Simulated linear and quadratic regression analyses were conducted to examine the relationships between MDA and Na+/K+, REL and MDA, yield and SPAD, yield and RWC, and yield and nitrogen. The Duncan multiple comparison method was utilized to assess the statistical significance of the difference (p<0.05). The results are presented as standard error (SE). Graphs were created using Origin 2021 software”

  • Materials and methods, Statistical analysis: What kind of regression analysis did the authors use?

Response: Thank you very much for your useful comments. This study investigate the relationship between MDA and Na+/K+, REL and MDA, yield and SPAD, as well as yield and RWC through a simulation of linearity. Additionally, the association between yield and nitrogen was analyzed using quadratic regression.

  • Results: It is not clear what statistical methods the authors used to identify the effects of straw nitrogen fertilizer on a particular parameter. An analysis of variance was used to test the difference between the means of the two samples "N: nitrogen fertilizer" and "S: straw"? It's not quite an effect, it's an observed mean difference tested with ANOVA. Again, the question arises why the parametric test was used. And it is not entirely clear what the size of each sample is. In M&M, it was stated that the total number of samples was 24. It turns out that each sample included 12? This is too little.

Response: Thank you to the reviewers for their valuable suggestions on my article. In this study, we employed a two-factor analysis of variance to examine the mutual influence of straw returning treatment and nitrogen application level on the measurement parameters. To evaluate the statistical significance of the differences, we used the Duncan multiple comparison method (p<0.05). The study conducted three replicates (n=3) for each treatment. The figures 2-10 in this paper represent the mean values of these three replicates. I apologize for any inconvenience caused to the reviewers and editors by not providing detailed descriptions of the statistical methods in the paper. We have now made the necessary modifications, Please refer to lines 186, 197, 236, 251, and 267-276 for specific modifications.

  • Column charts, which are predominantly used by the authors, represent mean values with an error of the mean. They do not reflect any of the effects of straw nitrogen fertilizer. Line graphs, perhaps filled with different colors, would look better.

Response:Thank you very much for youruseful comments.We have incorporated your suggestion and made the necessary modifications to the picture. The specific modifications are illustrated in figures 1-10.

  • Lines 612-614: the regression analysis did not include this, and it is not entirely correct to draw such a conclusion based on a three-year follow-up. Although intuitively this seems to be true, the results obtained are not enough for this conclusion.

Response: Thank you for your valuable comments. In response to your question, we have made corresponding modifications. The modifications are as follows: “Therefore the combination of straw returning and nitrogen fertilizer has been found to be beneficial for the growth of rice and the formation of its yield in saline-sodic land. And the effectiveness of this combination improves over time. However, regression analysis has shown that the optimal nitrogen application rate for rice varies depending on the number of years that straw has been returned to the field. It has also been observed that excessive nitrogen fertilizer application should be avoided when straw is returned to the field. This approach not only ensures stable rice yield in saline-sodi fields, but also has positive effects on the economic impact of fertilizer application and soil environment preservation.” → “Therefore, The combination of straw returning and nitrogen fertilizer has been found to be beneficial for the growth and yield of rice. It has been observed that the effectiveness of this combination improves with time. In this study, it was found that the best positive effect on yield was achieved when straw was combined with 250kg ha-1 nitrogen fertilizer. It is important to avoid excessive application of nitrogen fertilizer when straw is returned to the field. This method not only ensures the stability of rice yield in saline fields but also has positive effects on the economic aspects of fertilization and soil environmental protection.”

  • In the discussion, the authors often mention the effect on cell membranes. But this is based on data from other studies. When this is put forward as an assumption, it is good, but as a conclusion (lines 605-606) it is not correct - the authors did not conduct such an experiment.

Response:Thank you for your kindly suggestion. We have made the necessary revisions and the updated results can be found in line 647. We appreciate your valuable feedback.

  • Attention is drawn to the representation of the sources (references) used by the authors. It is worth expanding the range of sources, and include more from other regions.

Response: Our reference citations only consider content relevance, which provides us with an important hint. Consequently, we have made certain modifications, as indicated in lines 562-637.

Reviewer 2 Report

The problem of climate change affecting the entire globe and posing a threat to all crops. It is important to have such a study that allows you to meet the needs of users in the vicinity of the place of production, by applying product problems to interrupted deliveries of products, as is currently the case with products from Ukraine. Thus, it may not use environmentally neutral (250 kg of nitrogen fertilizer), but effective in creating a crop, it is the subject and guidelines for cultivation treatments.

The course of the experiment and its planning are correct, the statistical markings are correct. I propose to accept the work in its current form.

Author Response

Climate change is a global issue that threatens all crops. It is important to conduct such studies by applying product issues to disruptions in product delivery to meet the needs of users near the place of production, as is currently the case with products from Ukraine. Therefore, it may not use environmentally neutral (250 kg nitrogen fertilizer), but it can effectively create crops, it is the subject and guide of the cultivation treatment.

The experimental procedure and its planning are correct and the statistical markers are correct. I recommend accepting the job in its current form.

Response: We thank the reviewers for their positive feedback on our article. We are committed to continuous improvement and will remain committed to our work.

Reviewer 3 Report

Reviewer comments

Title

Looks good. However, it is a bit lengthy (23 words). Could it not be shortened

Abstract

This is a very lengthy summary. The lines 26 31 could be removed without affecting the abstract as an abstract

Line 33 sodic not sodi

Introduction

The line 113 is inappropriate. It is not at all clear what the definite article refers to? This sentence should start with something along the lines of ‘Previous studies (34) have shown….

Remove Therefore in line 120. In fact ‘the purpose of this…’ should form the basis of a new paragraph as it will round out the story. At the moment the rationale for the study is somewhat buried in the lengthy last paragraph

Materials and Methods

These seem appropriate and I am please to see that this is the result of a three year study

Results

This section is well written and the figures appropriate

Discussion

The discussion section is very clear. As noted below, there are far too many adverbs used in the text. These can all be removed without changing the point that the authors are trying to make

The English is good. The abstract is a bit lengthy and the introduction should be slightly rearranged in order to tell a story

the text is very wordy and could be tightened up throughout by removing the adverbs. One example amongst many is in lines 335-340

'However, tThere was no significant difference in the MDA  content and REL between the N2 and N3 treatments. Additionally, uUnder the same nitro- gen fertilizer level, both the RWC and Ψw increased with the increase in straw returning years. According to the results presented in Figure 6, shows it can be observed that the interaction  between S×Y has a noteworthy impact on the RWC and Ψw of rice grown in saline-sodic soil.'

Editing of this form should be applied throughout the text

Author Response

  • Looks good. However, it's a bit long (23 words). can't it be shortened

    Response: Thank you for your valuable suggestion. We have revised the title of the article. Please refer to Line 2-4 for details.

    This is a very lengthy summary. Lines 26-31 can be removed without affecting the summary as a summary

    Response: Thanks for your helpful comments. We have made the necessary modifications to this question. See lines 24-27 for details.

    Line 33 sodic not sodi

    Response: Thank you for your kind suggestion. I modified the text ("Saline-Sodium" → "Brine-Sodium"), and the modification is highlighted on line 29 of the text.

    Line 113 is inappropriate. Not sure what the definite article refers to at all? This sentence should start with "Previous research (34) has shown that...

    Response: Thanks for your helpful comments. We have made the necessary modifications to this question. See line 111 for details.

    Therefore, delete in line 120. In fact, "The purpose of doing this..." should form the basis of a new paragraph, as it will complete the story. For now, the rationale for this research is somewhat buried in the lengthy last paragraph.

    Response: Thank you for your kind suggestion. We have made the necessary modifications to this question. See line 119 for details.

    The Discussion section is very clear. As mentioned below, too many adverbs are used in the text. These can all be removed without changing the point the author is trying to make

    Response: Thank you for your affirmation and valuable suggestions on my article. We made changes to the removal of adverbs. See the Analysis and Discussion section in the Article Results for more information.

    The text is quite lengthy and can be tightened by removing adverbs. An example of this is lines 335-340

    Response: Thank you for your valuable suggestion. We have made the necessary revisions to this question. For more information, see lines 316 and 318, and the article Results Analysis section.